# JA-TN: Pick-and-Place Towel Shaping from Crumpled States based on TransporterNet with Joint-Probability Action Inference

**Halid Abdulrahim Kadi**
University of St Andrews
ah390@st-andrews.ac.uk

**Kasim Terzić**
University of St Andrews
kt54@st-andrews.ac.uk

**Abstract:** Towel manipulation is a crucial step towards more general cloth manipulation. However, folding a towel from an arbitrarily crumpled state and recovering from a failed folding step remain critical challenges in robotics. We propose joint-probability action inference JA-TN, as a way to improve TransporterNet's operational efficiency; to our knowledge, this is the first single data-driven policy to achieve various types of folding from most crumpled states. We present three benchmark domains with a set of shaping tasks and the corresponding oracle policies to facilitate the further development of the field. We also present a simulation-to-reality transfer procedure for vision-based deep learning controllers by processing and augmenting RGB and/or depth images. We also demonstrate JA-TN's ability to integrate with a real camera and a UR3e robot arm, showcasing the method's applicability to real-world tasks.

**Keywords:** Cloth Manipulation, Imitation Learning, Sim2Real Transfer

## 1 Introduction

Despite many recent efforts in the robotic community, cloth manipulation remains a key challenge due to its complex deformation and severe self-occlusion [1, 2]. Towel shaping is a set of critical subtasks of towel manipulation, including flattening and folding a rectangular fabric. Regarding initial states, folding can be classified into folding-from-flattened and folding-from-crumpled tasks, where the latter is a more complex problem [3] — a controller must flatten the towel first before folding it into a goal state; in addition, a robust controller should recover from failure states.

Most data-driven approaches train on a square fabric for flattening [4, 5, 6, 7, 8], folding-from-flattened [9, 10], or both [11, 12, 13, 14]. While some demonstrate their generalisability to other rectangular fabrics [4, 9, 12, 13], only a few can achieve an intended shaping on various types of fabrics from a large set of different initial states; they are based either on intermediate goal instructions [12, 13, 14] or a hierarchical system using hybrid action primitives [1, 15]. Our method JA-TN is the first to demonstrate various folding-from-crumpled tasks (see Figure 1b) using single-gripper pick-and-place (P&P) controllers on various rectangular fabrics varying in sizes, length-to-width ratios and colours. In a nutshell, we make the following contributions:

1. Our JA-TN towel-shaping controller extends the TransporterNet architecture [16] with our joint-probability action inference and achieves state-of-the-art performance for flattening, folding-from-flattened and folding-from-crumpled tasks in simulation.

2. We develop three standard benchmark domains — *Mono Square Fabric*, *Rainbow Square Fabrics*, and *Rainbow Rect Fabrics* — based on SoftGym [3] with towel-shaping tasks and corresponding oracle expert policies that are vital for the success of JA-TN.

8th Conference on Robot Learning (CoRL 2024), Munich, Germany.

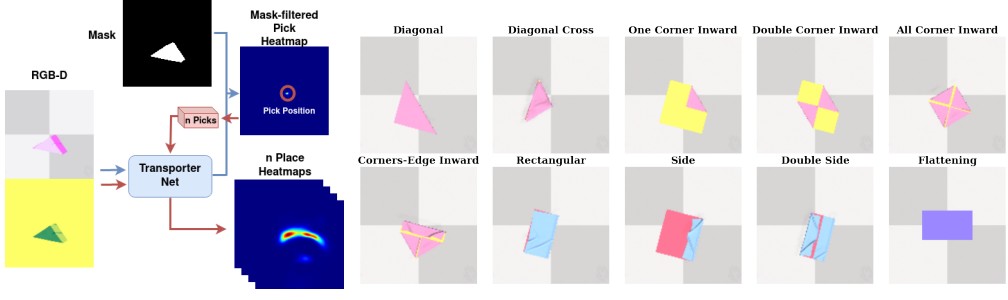

| (a) Action Inference | (b) Towel-shaping Task Variations |

Figure 1: JA-TN. Pick action inference takes RGB-D as input and uses a cloth mask to filter pick heatmap suggested by the transporter network (in blue), then it samples $n$ best pick positions to generate $n$ related place heatmaps (in red). Our method can complete all presented shaping tasks.

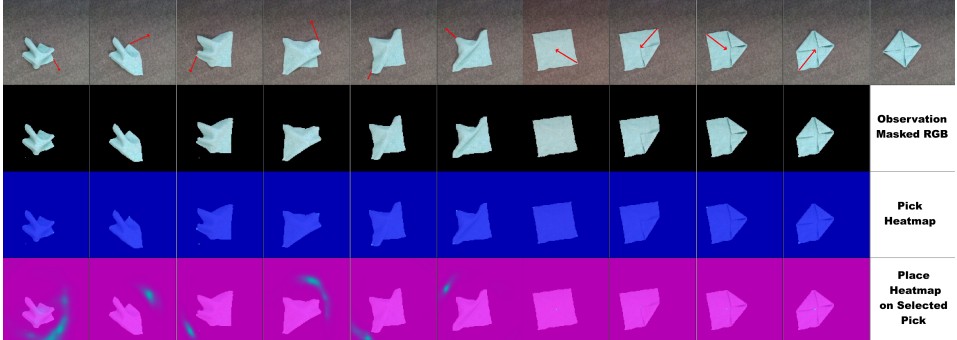

Figure 2: Qualitative Trajectory of Simulation-to-Reality Transfer of JA-TN

3. We propose a Sim2Real transfer strategy and a robot setup for vision-based data-driven controllers; our reality results demonstrate JA-TN's shaping ability in physical trials after training on masked-out RGB (and/or depth) images.

## 2 Related Work

**Robotic Cloth Manipulation**   Classical cloth manipulation systems are restricted to a narrow range of configurations, whereas data-driven approaches allow robust and general controls in open environments with a broader range of contextual configurations [2]. A large amount of data-driven research has tried to address cloth flattening and folding (including fabrics and garments) since 2018 [17, 18]. These methods mainly vary regarding their action space. (1) Gravity-based methods stretch the article in the air using dual grippers by picking the two key points [19] then adopt fling action [20] to flatten the article; finally, a folding heuristic method, such as *g-fold* [21], to achieve target shape. (2) Pick-and-place (-drag) methods employ a single or dual gripper to pick the key points on the article to transport them towards target locations for accomplishing the flattening and folding [22, 23, 9, 7]. (3) Adopting hybrid and variations of the above two action primitives demonstrated significant results in achieving many types of garment manipulation [1, 24, 18, 15].

**Data-driven P&P Towel Shaping**   Many data-driven controllers in P&P towel shaping utilise key-point detection [24, 25], template registrations [1], optical flow [26, 9], visual descriptors [27], point clouds [18] and mesh reconstructions [4, 28] as intermediate representations. Some others adopt end-to-end training [17], which produces a latent space that can be used for policy learning [22, 13, 6] and planning [23, 12, 7].

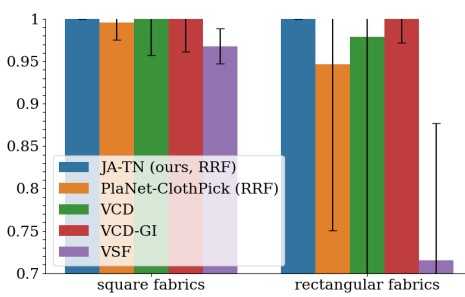
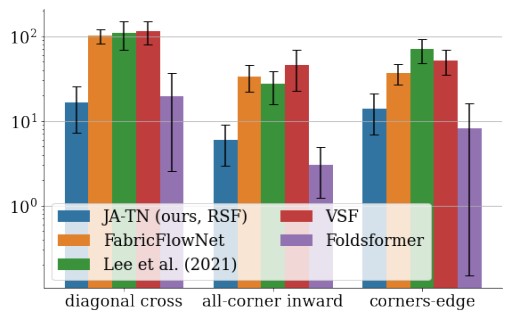

(a) Flattening, Normalised Improvement ↑     (b) Folding-From-Flattened, Mean Particle Distance ↓

Figure 3: Final-Step JA-TN (RGB-D w/o vision processing) Performance against Sate-of-The-Art Controllers; we evaluate in the settings of Lin et al. [4] (flattening) and Mo et al. [10] (folding-from-flattened); RSF and RRF denote that the method learns in our *Rainbow Square/Rectangular Fabrics* setting; the results of VCD, VCD-GI, VSF, FabricFlowNet, Lee et al. [13], VSF and Foldsformer are adopted from Lin et al. [4] and Mo et al. [10]. JA-TN achieves 100% success, 1.0 ± 0.0 of NC, 1.0 ± 0.01 of NI in flattening and matches SoTA performance in folding benchmarks, with mean particle distance (mm) of 16.40 ± 0.11, 6 ± 3.03, and 14.08 ± 7.22 for corresponding tasks.

Single-purpose deep reinforcement learning (RL) can flatten a crumpled towel [17, 7, 22, 4, 29] and perform specific folding-from-flattened tasks [29] with specifically engineered states and rewards biased towards the domain. Mesh-based planning methods [28], such as VCD and VCD-GI [4], and latent planning methods, such as PlaNet-ClothPick [7], achieve state-of-the-art (SoTA) performance on flattening a fabric. Goal-conditioned RL has struggled with general towel-shaping tasks due to the difficulty of engineering proper dense reward functions and learning from sparse ones [23, 12, 6]. Similar to our approach, Wu et al. [8] also utilise a TransporterNet to train a pick-and-place visual affordance map to flatten a square fabric without demonstration. Their method depends on pixel-wise reward function using domain-specific heuristics for self-supervised learning, and it only reaches 76% coverage in simulation and 68% coverage in real trials; it is not clear how it can be applied to various shapes of fabrics and folding tasks.

In contrast, successful approaches for folding adopt certain variations of imitation learning. Ganapathi et al. [11] use learned visual correspondences [30] across different fabric configurations to account for the spatial relationship between the current and the goal image; their system can perform a diverse range of fabric manipulation tasks, including various types of folding using a single demonstration trajectory as guidance. FabricFlowNet [9] achieves multi-step fabric folding from canonical states (flattened at the camera centre) with one or two grippers using goal-directed images; its underlying goal-condition policy learns to generate a pick heatmap conditioned on optical flows produced by a pre-trained flow network, then it selects the place action based on the optical flow and the chosen pick position, but it does not support folding from crumpled or arbitrary flattened states. Foldsformer [10] achieves multi-step folding-from-flattened by utilising a space-time attention mechanism [31] to capture the instruction information of demonstrations, but it cannot flatten. TransporterNet (TN) [16] is a vision-based end-to-end method that preserves the spatial structure of the RGB-D input to infer pick and place positions on images; Seita et al. [6] extend the approach to goal-conditioned control domains and apply it to cloth-shaping and bag manipulation tasks in simulation.

## 3 Method

**TransporterNet (TN)** is a vision-based end-to-end behaviour-cloning approach for P&P object manipulation [16]. It comprises of pick $q_{\text{pick}}(\cdot|\boldsymbol{x})$ and place $q_{\text{place}}(\cdot|\boldsymbol{x}, \boldsymbol{a}_{\text{pick}})$ networks where the latter is composed of convolutional key $f_{\text{key}}$ and query $f_{\text{query}}$ networks. TN learns to infer the pick-

action heatmap and selects the most probable action $\boldsymbol{a}_{\text{pick}} = \arg\max_{x,y} q_{\text{pick}}((x,y)|\boldsymbol{x})$, then uses the query-map of the region-of-interest of the pick action to cross-correlate to the key-map of the observation $\boldsymbol{x}$ for estimating its place action, i.e., $\boldsymbol{a}_{\text{place}} = \arg\max_{x,y} f_{\text{query}}(\boldsymbol{x}[\boldsymbol{a}_{\text{pick}}]) * f_{\text{key}}(\boldsymbol{x})$. Place network samples rotational instances of the region of interest (before feeding into the query network) to produce heatmap channels that signal the rotational degrees of the end-effector when placing. It adopts random-pivot rotation as an augmentation technique on RGB-D input images during training. The action maps use one-hot encoding with a value of 1 at the target pick position and 0 elsewhere. In inference, it selects the best combination of pick and place positions based on a selected pick position to act on the environment. Vanilla TN fails on fabric flattening tasks in Pybullet Raven Benchmarks [6, 16] and the simulation is far from realistic. However, TN shows promising results in other P&P domains so we assess and extend it in our towel-shaping tasks.

**Joint-Probability Action Inference**  To ensure an efficient selection of P&P positions during test time, we first mask the pick-heatmap with an object mask to filter out the pick positions that are not on the object, and we select the best joint probability of P&P, i.e.,

$$\boldsymbol{a}^* = \arg\max_{\boldsymbol{a}_{\text{pick}},\boldsymbol{a}_{\text{place}}} p_{\text{mask}}(\boldsymbol{a}_{\text{pick}}) \times q(\boldsymbol{a}_{\text{pick}}|\boldsymbol{x}) \times q(\boldsymbol{a}_{\text{place}}|\boldsymbol{a}_{\text{pick}},\boldsymbol{x}). \tag{1}$$

We sample 10 of the best picks to perform inference in practice and select the most probable one that differs from the last pick action. We utilise centre-pivot random rotation instead of random-pivot for data augmentation and update the network with a batch size of 10 (instead of the original 1). In the rest of the paper, we refer to this method as JA-TN. The success of JA-TN highly depends on high-quality expert trajectories, which we generate by manually designed oracle policies.

**Oracle Towel Smoothing (OTS)**  One-gripper P&P flattening, or smoothing, is a complex problem even if the true state of the article is known, and a controller should address different crumpled states of the towel. Table 1 presents the heuristics of our oracle algorithms in various scenarios.

*Flattening* case-policy picks and places the visible corner furthest from its target flattening position, which is calculated by scanning the whole pixel space to find the minimum total distance towards the targets from all cloth corners – considering symmetry including both sides and the vertical as well as horizontal displacements of the towel. *Revealing-Corner* policy picks the particle on the cloth-border and closest to a hidden corner and its target flattening position, then the policy drags the towel to the opposite of the hidden corner. *Revealing-Edge* policy targets the hidden edge-point furthest from its closest cloth border, then picks the cloth-border point and places it on the opposite side of the hidden point. *Untwisting* grabs the corner whose one cloth edge is above the other edge that excludes the corner and drags the grabbed point towards the perpendicular direction of the distance between existing twisting corners. *Dragging-to-Opposite* picks the cloth-border point furthest from the camera centre and drags it to the opposite side of the centre. Note that the OTS needs to use a more significant dragging distance for adjustment while addressing larger fabrics due to larger friction. The proposed algorithm is relatively robust regarding the fabric's initial crumpled states, size, and length-to-width ratio. Still, it may fail to address higher-order twisting and cases where a corner is wrapped inside the towel. Table 2 shows the policy can generate expert trajectories for behaviour cloning algorithms.

Table 1: Oracle Towel Smoothing (OTS). The oracle detects one of the 10 conditions in order and applies a particular scripted policy for each case.

| Case | Condition | Policy | Case | Condition | Policy |
|------|-----------|--------|------|-----------|--------|
| I | Succeeds to flatten | No Operation | VI | There is a level-1 twist | *Untwisting* |
| II | Fails to reveal a corner | *Revealing-Corner* | VII | There is an effective flattening | *Flattening* |
| III | Fails to reveal a corner twice | Follow the cases below | VIII | There is a hidden corner | *Revealing-Corner* |
| IV | Successfully reveals a corner | *Flattening* on the corner | IX | Almost flattened, but with hidden edge-points | *Revealing-Edge* |
| V | At the boundary of the view or small coverage | *Dragging-to-Opposite* | X | Otherwise | *Flattening* |

Table 2: Quantitative Results of Flattening Controllers. Our JA-TN shows superior performance and efficiency in both MSF and RRF domains compared to vanilla TN. Shown are Normalised Coverage (NC), Normalised Improvement (NI), steps-to-finish (#2F), and Success Rate (SR).

| Method | NC ↑ | NI ↑ | #2F ↓ | SR ↑ | NC ↑ | NI ↑ | #2F ↓ | SR ↑ |
|---|---|---|---|---|---|---|---|---|
| OTS | 1.0 ± 0.0 | 1.0 ± 0.0 | 12.49 ± 6.01 | 100.0 | 0.98 ± 0.06 | 0.99 ± 0.05 | 13.45 ± 9.53 | 96.43 |
| TN | 0.84 ± 0.2 | 0.72 ± 0.35 | 20.92 ± 10.12 | 52.31 | 0.5 ± 0.16 | 0.12 ± 0.2 | 27.41 ± 8.27 | 8.93 |
| JA-TN | **1.0 ± 0.02** | **0.99 ± 0.06** | **11.05 ± 6.09** | **98.46** | **0.96 ± 0.11** | **0.93 ± 0.22** | **12.61 ± 9.44** | **91.07** |

(a) Mono Square Fabric                                                              (b) Rainbow Rectangular Fabrics

**Oracle Towel Folding (OTF)**   Oracle policy for folding assumes a flattened article and maps its surface to a 2-dimensional plane ranging from 0 to 1 — (0, 0) represents the top-left particle of the towel whilst (1,1) denotes the bottom-right. OTF regards a specific folding as a sequence of P&P positions on the article map.

**Training Details of JA-TN and TN**   We train JA-TN and TN using 2000 OTS demonstrations for flattening, 1000 OTF demonstrations for corresponding folding-from-flattened tasks and the combination of both for folding-from-crumpled tasks. We train JA-TN with 80,000 update steps and vanilla TN with 40,000 update steps as it starts to overfit beyond this (see Figure 9b and 9c). We also make the following modifications to adapt the vanilla TN to our shaping tasks. We remove rotation-variant sampling because it is not helpful in our domain. We use image resolution $128 \times 128$ instead of the original $320 \times 160$. Rather than synthesising the top view from 3 separate cameras as in the original paper, we use a single camera to capture the top view of the towel. Moreover, we clip the depth image between 1.47 and 1.51 (as the camera is 1.5m above the surface in simulation) and normalise both RGB and depth images to range from -0.5 to 0.5.

## 4  Experiments

We implement our methods in SoftGym [3], which is a particle-based cloth simulator, and we develop three domains: (1) *Mono Square Fabric* (MSF) contains a 40cm × 40cm square fabric that has the same back and front colours across the trials; (2) *Rainbow Square Fabrics* (RSF) contains a set of square fabrics whose side-lengths are uniformly sampled from 20cm to 70cm, and the colours of this fabric are also uniformly sampled from all possible colours; (3) *Rainbow Rectangular Fabrics* (RRF) is a similar setting to RSF but the length and the width of a fabric can be different. All three domains include a top-down camera that looks down at the centre of the world surface from a height of 1.5m. Cloth stretch, bend, and shear stiffness are set to (0.8, 1, 0.9) with a particle radius of 6.25mm and a particle mass of 0.5.

**Towel Flattening**   We employ bounded (clipped between 0 and 1) normalised coverage (NC) and normalised improvement (NI) to evaluate the step-wise performance of flattening controllers [4]. We additionally use success rate (SR) and steps-to-finish (#2F), on success or end of the trial, to assess the robustness and efficiency of the methods; a flattening trial succeeds when the controller can achieve above 0.99 of NC within 30 steps. We evaluate the controllers on 65 trials of MSF and 56 trials of RRF. Figure 3a shows the final step performance of JA-TN against SoTA controllers in the evaluation setting of Lin et al. [4]. Table 2 shows improvement of our JA-TN compared to vanilla TN in terms of performance and efficiency in both MSF and RRF domains.

**Towel Folding from a Flattened State**   For folding-from-flattened tasks, we use standard mean particle distance (MPD) between the particle pairs in possible corresponding folding groups [9] as well as additional largest particle distance (LPD) to evaluate the step-wise performance. A folding succeeds when LPD goes below a predefined threshold value (7cm for diagonal-cross folding and 4.5cm for others) within 25 steps. All folding-from-flattened tasks in the 3 domains share 30 test trials where the article is randomly rotated and placed within the camera view. We train JA-TN

Table 3: Quantitative Results of Folding-from-Flattened Controllers compared to vanilla TN. LPD and MPD are shown in mm for the average over all suitable foldings. JA-TN shows consistent superior performance in all domains. Although the oracle has access to a full simulation state, JA-TN outperforms it in some cases based on RGBD input alone.

| Method | LPD ↓ | MPD ↓ | SR ↑ | LPD ↓ | MPD ↓ | SR ↑ | LPD ↓ | MPD ↓ | SR ↑ |
|---|---|---|---|---|---|---|---|---|---|
| OTF | 20.79 ± 6.87 | 10.36 ± 3.48 | 98.89 | 40.2 ± 20.47 | 17.78 ± 7.99 | 90.0 | 58.26 ± 80.16 | 29.26 ± 41.01 | 87.78 |
| TN | 34.15 ± 23.15 | 15.11 ± 9.94 | 89.17 | 77.34 ± 41.64 | 25.55 ± 11.45 | 61.11 | 130.32 ± 100.15 | 55.08 ± 39.2 | 31.11 |
| JA-TN | **30.19 ± 19.73** | **12.43 ± 6.55** | **93.67** | **40.83 ± 23.29** | **17.43 ± 7.55** | **88.33** | **44.32 ± 41.48** | **22.24 ± 17.63** | **90.0** |
| | (a) Mono Square Fabric | | | (b) Rainbow Square Fabrics | | | (c) Rainbow Rectangular Fabrics | | |

Table 4: Quantitative Results of Folding-from-Crumpled Controllers

| Method | LPD ↓ | MPD ↓ | #2F ↓ | SR ↑ | LPD ↓ | MPD ↓ | #2F ↓ | SR ↑ |
|---|---|---|---|---|---|---|---|---|
| Oracle | 31.12 ± 20.28 | 12.88 ± 6.87 | 19.59 ± 10.02 | 95.69 | 55.57 ± 64.62 | 24.12 ± 26.86 | 32.13 ± 15.79 | 73.21 |
| JA-TN | 37.37 ± 29.31 | 15.05 ± 10.4 | 24.13 ± 12.07 | 94.46 | 114.76 ± 107.64 | 49.11 ± 45.69 | 42.01 ± 15.66 | 50.89 |
| | (a) Mono Square Fabric | | | | (b) Rainbow Square Fabrics | | | |

and TN on 1,000 demonstrations for each folding and examine (1) all types of folding illustrated in Figure 1b on MSF, (2) diagonal-cross, all-corner inward, corners-edge inward and rectangular folding in RSF, and (3) side, double-side and rectangular foldings in RRF. We did not examine double-straight folding [10] as it is challenging even for the oracle to achieve with a single gripper. Table 3 and Figure 3b show that JA-TN performs robustly on these folding tasks in all 3 domains, and it matches the SoTA performance of Foldsformer [10], a method that cannot fold from crumpled.

**Towel Folding from Crumpled** Folding-from-crumpled tasks share the same initial states as the ones for flattening and the same success criteria as folding-from-flattened tasks, but they must finish within 55 steps. We train JA-TN on 2,000 flattening + 1,000 folding-from-flattened demonstrations. We examine (1) all-corner inward, diagonal cross, corners-edge inward, double-side and rectangular foldings in MSF and (2) the latter in RRF. Table 4 shows JA-TN's high performance in the MSF setting and its potential in the more complex RRF setting.

**Ablation Study** We conduct ablation tests of JA-TN on flattening and double-side folding tasks in RRF. Table 5 shows that mask filtering is essential for the success of flattening agent as `No Mask` ablation tends to choose the pick position of `No-Op` policy, whilst the joint-action inference is crucial for the success of folding agent as it reduces ineffective folding steps. Training with a larger batch size and using centre-pivot rotation in data augmentation are critical for the system's success in both tasks, as the former reduces overfitting and improves the learning (see Figure 9b and 9c), and the latter avoids feeding the networks with undesirable actions.

**Sim2Real Transfer** Many data-driven P&P towel-shaping methods developed in SoftGym demonstrate the ability to transfer to real trials mostly using depth-only images [4, 10, 9, 8]. We re-train JA-TN with masked RGB (and/or depth) images in simulation and adopt a `segment-anything` model [32] with heuristic mask selection in physical inference – the largest average colour difference between the masked and background regions. We employ a top-down RealSense D435i camera positioned 0.72 metres above the table surface and rely on human judgment to assess the success of a trial. We choose one square (sky-blue, 14cm) and one rectangular (white, 15cm×7cm) fabric for physical evaluations. During the physical experiment, we feed RGB(-D) camera input to JA-TN and ask it to generate pick-and-place action primitives (see Figure 4b). To isolate the effect of grasping errors common in this domain such as misgrasping and grasping multiple layers, we then manually perform each action on the real cloth (human in the loop) before repeating the process. This eliminates grasping errors and allows us to focus on the quality of the learned policy operating

Table 5: JA-TN's Ablation study on RSF task. `Batch Size 1` - network trains with one observation-action pair per update, similar to the original TN; `Random Pivot` - rotational augmentation using a randomly sampled pixel on the image, as in the original TN; `No MJ` - without mask-filtered joint-probability action inference; `No Mask Filter` - without filtering pick heatmap with cloth-mask; `No Joint` - no joint-probability action inference. All additional components of the JA-TN are crucial for the system's success in folding and flattening.

| Ablation | NC ↑ | NI ↑ | #2F ↓ | SR ↑ | LPD ↓ | MPD ↓ | SR ↑ |
|---|---|---|---|---|---|---|---|
| Batch Size 1 | 0.73 ± 0.19 | 0.48 ± 0.37 | 22.52 ± 11.87 | 30.36 | 95.15 ± 90.18 | 33.97 ± 30.86 | 46.67 |
| Random Pivot | 0.79 ± 0.22 | 0.63 ± 0.4 | 20.79 ± 11.39 | 48.21 | 67.76 ± 76.05 | 26.88 ± 26.71 | 70.0 |
| No MJ | 0.5 ± 0.21 | 0.24 ± 0.32 | 29.0 ± 5.29 | 3.45 | 59.83 ± 71.8 | 21.83 ± 20.14 | 83.33 |
| No Mask Filter | 0.61 ± 0.25 | 0.13 ± 0.3 | 27.41 ± 8.27 | 8.93 | 62.11 ± 76.74 | 23.23 ± 24.76 | 80.0 |
| No Joint | 0.92 ± 0.17 | 0.85 ± 0.32 | 13.64 ± 10.99 | 78.57 | 86.8 ± 94.37 | 29.65 ± 28.64 | 66.67 |
| JA-TN | **0.96 ± 0.11** | **0.93 ± 0.22** | **12.61 ± 9.44** | **91.07** | **45.12 ± 43.92** | **20.01 ± 15.09** | **90.0** |
| | (a) Flattening | | | | (b) Double-Side Folding from Flattened | | |

Table 6: Success Rate of Masked-RGB JA-TN planning on Real Fabrics. (sqr) denotes square fabric, and (rect) denotes rectangular fabric.

| initial states | flattening (sqr) | diagonal-cross (sqr) | corners-edge inward (sqr) | all-corner inward (sqr) | double-side (sqr) | rectangular (sqr) | flattening (rect) | double-side (rect) | rectangular (rect) |
|---|---|---|---|---|---|---|---|---|---|
| flattened | N/A | 5/5 | 5/5 | 5/5 | 3/5 | 1/5 | N/A | 4/5 | 2/5 |
| crumpled | 7/10 | 4/5 | 3/5 | 4/5 | 2/5 | 0/5 | 10/10 | 2/5 | 1/5 |

on live RGB-D images. This demonstrates that our simulation-trained system can create effective policies on real towels based on real camera input (see Figure 2). Whilst flattening and folding-from-flattened systems transfer nicely, flattening-from-crumpled agents cannot directly transfer; we must stitch the flattening and folding controllers with a predefined mask area. Table 6 shows our method's plausibility in physical trials on most tasks using RGB images, but note that it struggles to perform double-side and rectangular folding. We have also conducted experiments using a UR3e robot arm and an Active Parallel Gripper with a printed stick-like extension, which have managed to successfully flatten real fabrics in a completely autonomous fashion (see Figure 4c).

## 5   Discussion

JA-TN can effectively flatten various fabrics from highly crumpled states in both simulation and physical applications, regardless of their colour, size, length-to-width ratio, or background. It can also perform different folding types from flattened and crumpled states. To our knowledge, this is the first learning algorithm capable of excelling in all three tasks. It has potential applications in other P&P domains and object shapes, as we did not introduce inductive biases into the learning process apart from the Oracle policies — Figure 10 in the Appendix demonstrates that JA-TN can be applied to other complex shapes of cloth-like deformable objects, such as T-shirts.

Behaviour cloning methods like JA-TN, trained with around 20,000 transitional steps, are much more data-efficient compared to reinforcement learning (RL) methods, such as PlaNet-ClothPick (around 1 million steps) [2] and VCD (around 100,000 steps) [4] for fabric flattening. For folding, it is hard to engineer reward functions for each type; Lee et al. [13] (using Hindsight Replay Buffer [33]) and VSF [12] (using visual MPC planning [34]) adopt a goal-condition strategy to bypass the problem, but these systems do not outperform other supervised learning and imitation learning methods such as FabricFlowNet [15], Foldsformer [10] and our JA-TN. To our knowledge, no RL algorithm has demonstrated the ability to fold directly from a crumpled state.

**Limitations**   JA-TN struggles with highly wrapped and severely twisted states in simulations and folded states (both outside and inside) in reality due to its inability to distinguish folds when both sides are the same colour. While it reliably performs most folding-from-flattened tasks, we see room

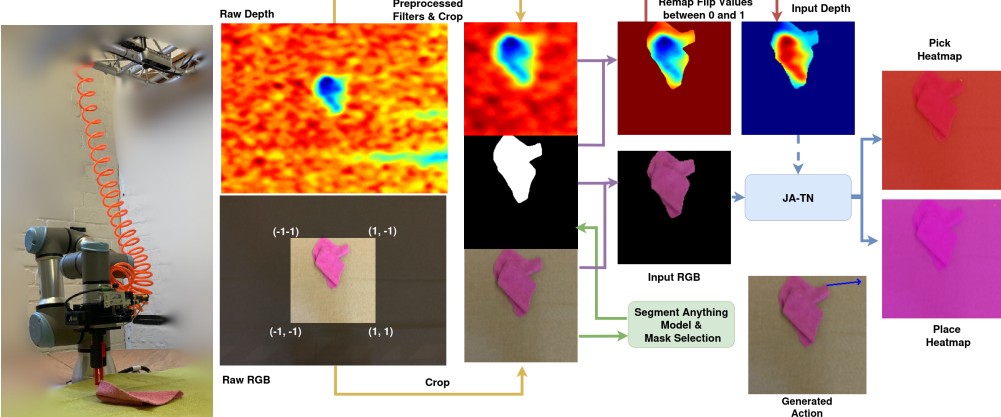

(a) Robot Setup     (b) Real Vision Preprocessing & Action Generation

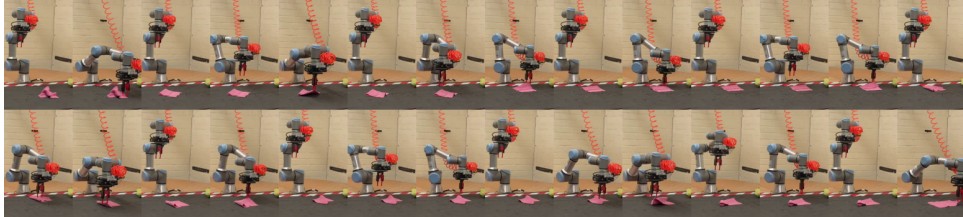

(c) A Qualitative Demonstration of Robot Fabric Flattening

Figure 4: Sim2Real Vision Processing Procedure. Real-depth images often suffer from noise, blurriness, and holes, even after calibration and filtering. To address this, we apply a cloth mask, heuristically selected from a segmentation network (green line), and set the depth image background to the camera height (purple line). Since fabrics vary in thickness, we standardise the cloth-top and cloth-bottom to 0 and 1 for robustness, then invert these values so 0 represents the background and deepest view, while 1 represents the cloth top (red line). We also normalise RGB images and apply mask segmentation to RGB images, ensuring the policy remains background-agnostic (purple line).

for improvement by addressing rectangular folding in simulations and reality as key picking points are often wrapped inside during the process; the system cannot distinguish when one side of the fabric overlaps at the bottom but still needs completion, so it may require a goal configuration to improve its recognition capabilities. Lastly, JA-TN only transitions from flattening to folding when it perceives the fabric as perfectly flattened, making physical trials difficult.

**Future work**    We aim to address the identified limitations of the shaping controller and enhance the transferability of JA-TN in folding from a crumpled state. Additionally, we plan to extend JA-TN's capabilities in two-gripper cloth manipulation to achieve more effective rectangular and double-straight foldings. We also intend to evaluate our method's effectiveness in other garment-shaping tasks. Furthermore, we seek to use goal-conditioned methods to unify various folding techniques, eliminating the need for training specialised systems for each folding type.

# 6   Conclusion

We propose a Joint Probability TransporterNet (JA-TN) that achieves state-of-the-art performance for flattening and folding-from-flattened tasks. We believe JA-TN can be applied to other pick-and-place domains as we do not introduce inductive biases towards our domain apart from the Oracle expert scripts. Our experiments on real towels and real camera images show our method's promise in real-world towel-shaping trials, and we plan to improve the performance of folding-from-crumpled controllers and their transferability.

**Acknowledgments**

We would like to thank Dr Stuart Norcross for setting up GPU machines for our use. We also wish to express our gratitude to Mr. Nicol Thomson at the University of St Andrews for setting up the UR3e robot and printing the stick-like gripper extension for robotic demonstrations. Additionally, we are grateful to the anonymous reviewers for their valuable feedback and suggestions that helped improve this paper.

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

# 7 Appendix

## 7.1 Initialisation Script for Crumpled States in Simulation

We run 1000 trials for Mono Square Fabric (MSF), 2000 for Rainbow Square Fabrics (RSF), and 5000 for Rainbow Rectangular Fabrics (RRF), where the trial's configuration ID seeds the random state. Crumpled states are initialised as follows: (1) the script selects a particle on the fabric and lifts it to a random height between 0 and 40cm; (2) after it stabilises in the air, the cloth is dropped; then (3) the script relocates the fabric (without changing the crumpled state) to a random position within a 30cm circle centred around the camera; (4) then, 70% of the time, it again selects a random particle and drags the cloth towards a random position in a circle with a radius of 30cm and a height of 10cm; it drops the cloth and re-centres it randomly again; finally, (5) the cloth's re-centring and random rotation continue until all particles are within the camera view. This results in a wide range of complex initial states.

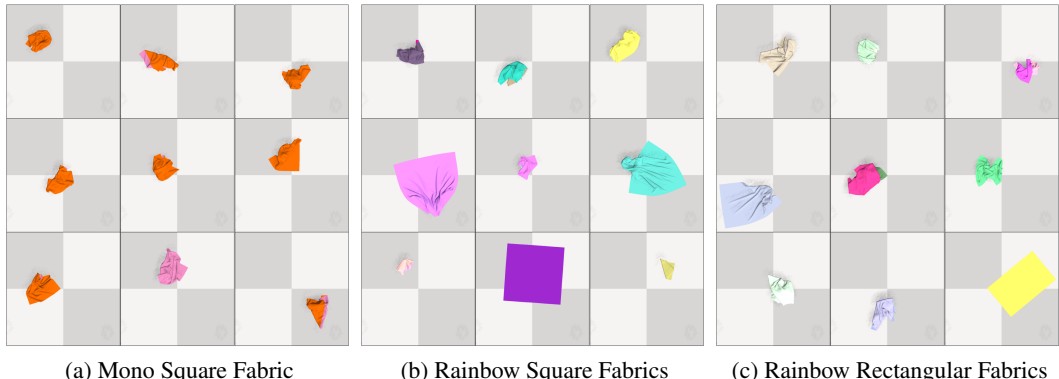

| (a) Mono Square Fabric | (b) Rainbow Square Fabrics | (c) Rainbow Rectangular Fabrics |

Figure 5: Crumpled Initial States for flattening and folding-from-flattened tasks

## 7.2 More Details on Oracle Policies

**Heuristics for OTF**    Rectangular, side, and double-side foldings are more challenging to achieve with one-picker operations, as they tend to conceal key pick-points inside the article, leading to failure; hence, OTF carefully produces the folding sequence based on fabric properties, which avoids such scenarios and the overlapping of the two sides in double-side folding.

**Limitations of Oracles**    As shown in Figure 6, the flattening Oracle struggles with highly twisted and wrapped states, but this only occurs once in all 56 tested trials in simulation. Figure 7 demonstrates that the folding Oracle cannot detect failing states until reaching designated folding steps, and rectangular and double-side foldings are particularly challenging due to the concealed key points.

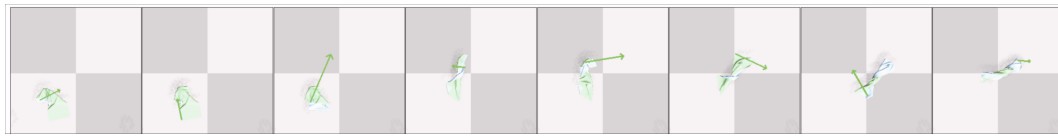

Figure 6: A Flattening Failure of Oracle Towel Smoothing

## 7.3 More Details on Sim2Real Experiments

**Preprocessing of Real Images**    As shown in Figure 4b, the real depth image can be noisy and blurry, and it can also contain holes even after calibration and being applied depth pre-processing

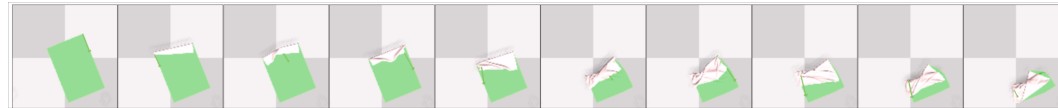

Figure 7: A Rectangular-Folding Failure of Oracle Towel Folding

filters provided by `Realsense` SDK. To address this, we apply a cloth mask and make the background of the depth images the same as the camera height. However, different fabrics have different thicknesses, so we map the cloth-top and cloth-bottom consistently to 0 and 1 to make our system robust to cloth thickness. Then, we flip the values so that 0 represents the background and the deepest part of the view, and 1 represents the cloth top. We also map the values to 0 and 1 for RGB images by dividing 255 from the original RGB image. Then, we applied mask segmentation on depth and RGB images so that the policy can be agnostic to the view background, a common practice in this domain [9, 10].

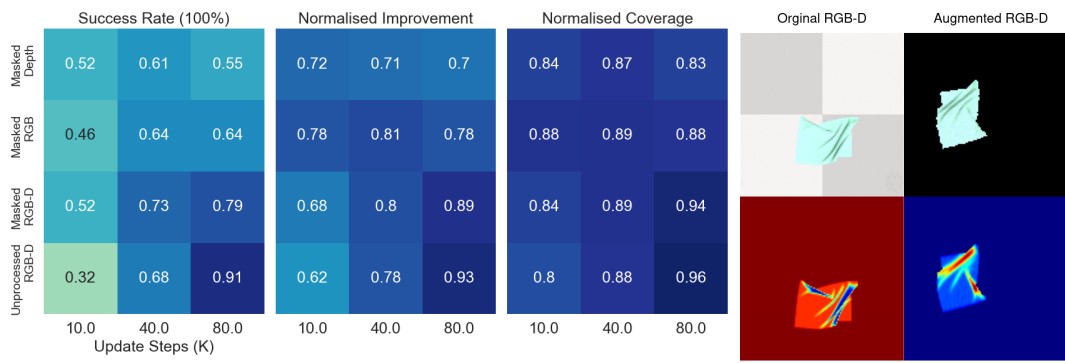

(a) Flattening Performance of Sim2Real Policies in RRF    (b) Vision Processing for Training

Figure 8: Sim2Real Learning Policies

**Training details of Sim2Real Policies**   To transfer the learned policy into the real world, apart from remapping, value flipping and masking as in preprocessing real-depth images, we also add Gaussian noise of 0.01 and Gaussian blurring with a kernel of 11, during training, to the simulated depth to emulate the real depth images (see Figure 8b). We also add a Gaussian noise of 0.02 to the simulated RGB images. Figure 8a shows how these augmentations influence JA-TN performance on flattening.

**Robot Setup**   As shown in Figure 4a, we use a 6-degree-of-freedom UR3e robot arm, an Active Parallel gripper and a Realsense D435i camera to conduct our robotic experiments. The camera is 0.72 meters above the table surface; we manually adjust the camera pose so that the depth images give an even depth reading across the view, and the camera is parallel to the robot base axes. We also 3D-printed a stick-like thin gripper extension to grasp the target cloth point with higher precision; whilst open, the tip-to-tip distance is 2.4cm. The robot working space is a $28cm \times 28cm$ square region on table syrface. We adopt MoveIt 2's Python API under ROS 2 Humble [35] with OMPL's `RRTConnect` [36] motion planner to achieve quasi-static pick-and-place robot control as well as establishing the communication between the robot and GPU program.

The transformation between the normalised pixel space (see the raw RGB image in Figure 4b) and the world space (same as robot base space) is calculated using the cropping offset and scale of work space, the intrinsic of the camera and the extrinsic transformation matrix between the camera frame and the robot's base frame. The calibration of the extrinsic parameters can be carried by eye-to-hand calibration, but it does not necessary gives a precise solution; hence a further fine-tuning is required.

For each pick-and-place action, the end-effector goes as deep as it can on the table to grasp the cloth (i.e., the tip of the end-effector is at the same height as the robot's baseline). Then, it picks the cloth up 5cm above the surface to go to the place point that is 3cm above the surface to release the cloth. While taking each picture, the robot gets to a home position that makes it completely absent from the camera view.

**Limitation of the Robotic System**  The gripper can misgrasp the cloth, grasp multiple layers of the cloth, and the cloth can attached to the gripper after release; the solutions for tackling these challenges are out of the scope of this paper. Also, we encounter edge (and/or corner) alignment issues during folding tasks. Currently, planning each motion trajectory takes 2 seconds, and we plan to explore other motion planners to improve efficiency. Additionally, the robot's workspace is limited because the planned trajectory fails to produce smooth joint movements when it approaches its base. Besides, the current robotic system cannot automatically detect the success states, and it only stops when it reaches defined maximum steps. Lastly, we must manually run the associated configuration for an intended task.

## 7.4 Supplementary Results

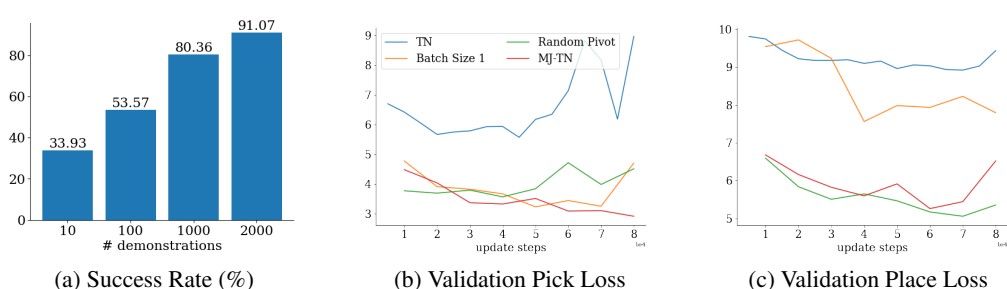

(a) Success Rate (%)    (b) Validation Pick Loss    (c) Validation Place Loss

Figure 9: Comparison JA-TN's Ablations on Flattening in RSF

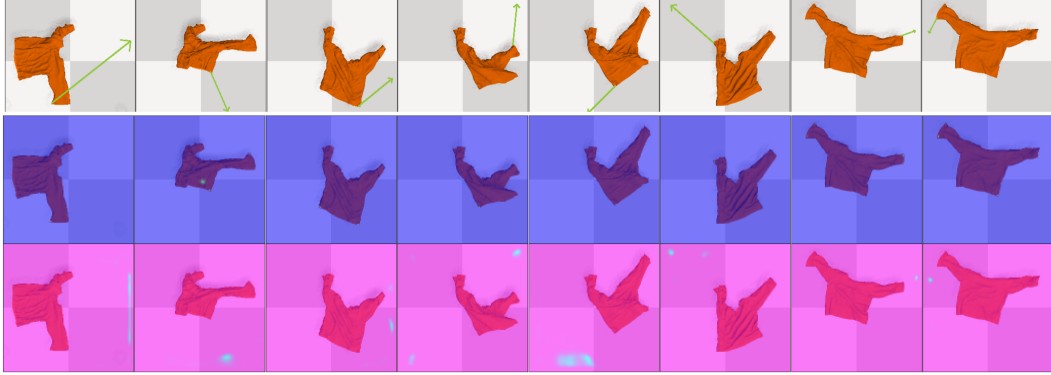

Figure 10: Qualitative JA-TN's Performance on Tshirt Flattening in Simulation. With only 20 human demonstrations, it can reach 70% normalised coverage across 30 test trials.

Table 7: Task-wise Quantitative Results of Folding-from-Flattened Controllers

| Method | LPD ↓ | MPD ↓ | SR ↑ | LPD ↓ | MPD ↓ | SR ↑ | LPD ↓ | MPD ↓ | SR ↑ |
|---|---|---|---|---|---|---|---|---|---|
| OTF | 8.25 ± 3.3 | 6.61 ± 2.75 | 100.0 | 8.6 ± 3.5 | 6.44 ± 2.7 | 100.0 | 4.86 ± 5.83 | 10.51 ± 5.17 | 100.0 |
| TN | 10.34 ± 5.54 | 8.55 ± 5.57 | 100.0 | 9.9 ± 5.33 | 6.58 ± 4.02 | 100.0 | 14.87 ± 7.25 | 8.15 ± 3.87 | 93.33 |
| JA-TN | 15.55 ± 8.65 | 12.93 ± 7.98 | 90.0 | 10.41 ± 5.16 | 8.0 ± 4.65 | 100.0 | 16.79 ± 7.14 | 9.19 ± 3.76 | 100.0 |

(a) Diagonal Folding in MSF  (b) One-Corner Inward in MSF  (c) Double-Corner in MSF

| Method | LPD ↓ | MPD ↓ | SR ↑ | LPD ↓ | MPD ↓ | SR ↑ |
|---|---|---|---|---|---|---|
| OTF | 39.98 ± 7.39 | 18.48 ± 3.36 | 100.0 | 56.9 ± 17.8 | 22.11 ± 6.87 | 90.0 |
| TN | 48.37 ± 12.32 | 20.39 ± 4.78 | 100.0 | 66.22 ± 23.18 | 28.08 ± 10.58 | 80.0 |
| JA-TN | 47.43 ± 9.14 | 16.92 ± 3.4 | 100.0 | 62.18 ± 23.17 | 24.07 ± 9.98 | 83.33 |

(d) Diagonal-Cross in MSF  (e) Diagonal-Cross in RSF

| Method | LPD ↓ | MPD ↓ | SR ↑ | LPD ↓ | MPD ↓ | SR ↑ |
|---|---|---|---|---|---|---|
| OTF | 16.86 ± 4.04 | 8.71 ± 2.22 | 100.0 | 21.8 ± 7.84 | 9.54 ± 3.0 | 100.0 |
| TN | 15.74 ± 4.01 | 7.37 ± 2.45 | 100.0 | 29.95 ± 30.24 | 10.05 ± 6.81 | 90.0 |
| JA-TN | 19.22 ± 6.49 | 8.18 ± 2.88 | 93.33 | 27.7 ± 35.07 | 9.88 ± 4.84 | 96.67 |

(f) All-Corner Inward in MSF  (g) All-Corner Inward in RSF

| Method | LPD ↓ | MPD ↓ | SR ↑ | LPD ↓ | MPD ↓ | SR ↑ |
|---|---|---|---|---|---|---|
| OTF | 25.97 ± 8.8 | 9.61 ± 3.77 | 96.67 | 32.26 ± 15.26 | 11.33 ± 4.53 | 93.33 |
| TN | 38.0 ± 15.92 | 13.35 ± 4.92 | 83.33 | 135.86 ± 71.5 | 38.52 ± 16.97 | 13.33 |
| JA-TN | 48.17 ± 42.53 | 13.6 ± 6.88 | 81.67 | 36.26 ± 19.56 | 14.25 ± 6.76 | 86.67 |

(h) Corners-Edge Inward in MSF  (i) Corners-Edge Inward in RSF

| Method | LPD ↓ | MPD ↓ | SR ↑ | LPD ↓ | MPD ↓ | SR ↑ |
|---|---|---|---|---|---|---|
| OTF | 22.91 ± 5.63 | 8.35 ± 2.08 | 100.0 | 43.31 ± 57.32 | 17.78 ± 23.37 | 93.33 |
| TN | 67.01 ± 56.19 | 24.2 ± 17.47 | 66.67 | 173.49 ± 104.72 | 63.03 ± 31.64 | 10.0 |
| JA-TN | 35.7 ± 25.99 | 11.88 ± 6.45 | 96.67 | 45.12 ± 43.92 | 20.01 ± 15.09 | 90.0 |

(j) Double-Side in MSF  (k) Double-Side in RRF

| Method | LPD ↓ | MPD ↓ | SR ↑ | LPD ↓ | MPD ↓ | SR ↑ |
|---|---|---|---|---|---|---|
| OTF | 22.68 ± 9.34 | 8.46 ± 3.45 | 100.0 | 42.36 ± 41.16 | 23.04 ± 20.22 | 93.33 |
| TN | 27.79 ± 11.08 | 12.59 ± 5.9 | 100.0 | 97.01 ± 78.89 | 45.44 ± 29.28 | 36.67 |
| JA-TN | 26.95 ± 9.54 | 12.57 ± 4.83 | 100.0 | 34.17 ± 19.99 | 18.01 ± 7.92 | 96.67 |

(l) Side in MSF  (m) Side in RRF

| Method | LPD ↓ | MPD ↓ | SR ↑ | LPD ↓ | MPD ↓ | SR ↑ | LPD ↓ | MPD ↓ | SR ↑ |
|---|---|---|---|---|---|---|---|---|---|
| OTF | 26.98 ± 14.04 | 16.06 ± 5.86 | 93.33 | 49.84 ± 40.96 | 28.14 ± 17.57 | 76.67 | 89.12 ± 142.01 | 46.96 ± 79.43 | 76.67 |
| TN | 68.93 ± 78.66 | 32.31 ± 36.4 | 70.0 | 97.01 ± 78.89 | 45.44 ± 29.28 | 36.67 | 137.53 ± 118.61 | 60.17 ± 50.37 | 33.33 |
| JA-TN | 33.54 ± 40.19 | 17.47 ± 17.81 | 93.33 | 37.16 ± 15.37 | 21.53 ± 8.6 | 86.67 | 53.68 ± 60.51 | 28.7 ± 29.88 | 83.33 |

(n) Rectangular in MSF  (o) Rectangular in RSF  (p) Rectangular in RRF

Figure 11: Qualitative Trajectory of Double-Side Folding from Crumpled of JA-TN

Table 8: Task-wise Quantitative Results of Folding-from-Crumpled Controllers

| Method | LPD ↓ | MPD ↓ | #2F ↓ | SR ↑ | LPD ↓ | MPD ↓ | #2F ↓ | SR ↑ |
|---|---|---|---|---|---|---|---|---|
| Oracle | 30.56 ± 33.82 | 10.83 ± 11.29 | 25.22 ± 11.47 | 93.85 | 54.53 ± 73.66 | 19.96 ± 26.21 | 31.23 ± 16.17 | 71.43 |
| JA-TN | 36.48 ± 28.52 | 14.13 ± 10.85 | 28.06 ± 12.47 | 93.85 | 105.13 ± 94.99 | 39.4 ± 33.73 | 41.41 ± 16.12 | 51.79 |

(a) Double-Side in MSF          (b) Double-Side in RSF

| Method | LPD ↓ | MPD ↓ | #2F ↓ | SR ↑ | LPD ↓ | MPD ↓ | #2F ↓ | SR ↑ |
|---|---|---|---|---|---|---|---|---|
| Oracle | 30.26 ± 20.81 | 16.32 ± 7.56 | 23.14 ± 12.61 | 89.23 | 56.61 ± 55.58 | 28.28 ± 27.51 | 33.04 ± 15.4 | 75.0 |
| JA-TN | 45.81 ± 45.27 | 21.75 ± 20.97 | 30.35 ± 14.23 | 86.15 | 124.4 ± 120.28 | 58.82 ± 57.65 | 42.61 ± 15.2 | 50.0 |

(c) Rectangular in MSF          (d) Rectangular in RSF

| Method | LPD ↓ | MPD ↓ | #2F ↓ | SR ↑ | LPD ↓ | MPD ↓ | #2F ↓ | SR ↑ |
|---|---|---|---|---|---|---|---|---|
| Oracle | 48.34 ± 25.41 | 20.45 ± 8.0 | 14.66 ± 9.45 | 98.46 | 16.9 ± 7.63 | 7.71 ± 2.61 | 17.34 ± 8.29 | 98.46 |
| JA-TN | 43.33 ± 8.9 | 16.86 ± 3.86 | 16.78 ± 10.61 | 100.0 | 26.27 ± 37.26 | 9.96 ± 7.93 | 21.35 ± 11.59 | 95.38 |

(e) Diagonal-Cross in MSF          (f) All-Corner Inward in MSF

| Method | LPD ↓ | MPD ↓ | #2F ↓ | SR ↑ |
|---|---|---|---|---|
| Oracle | 29.55 ± 13.74 | 9.09 ± 4.86 | 17.62 ± 8.27 | 98.46 |
| JA-TN | 34.98 ± 26.61 | 12.56 ± 8.39 | 24.11 ± 11.45 | 96.92 |

(g) Corners-Edge Inward in MSF

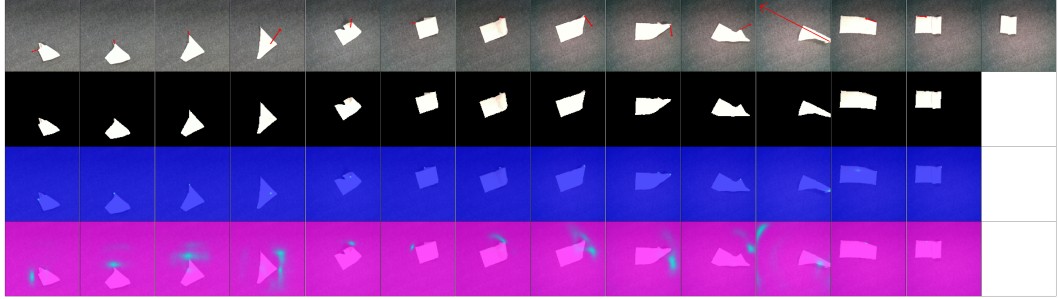

Figure 12: Qualitative Result of Double-side Folding in Reality

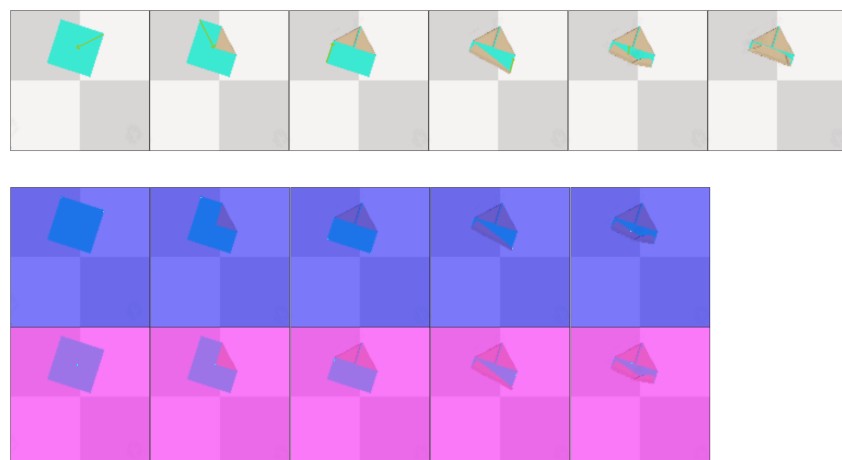

Figure 13: Qualitative Corners-Edge Inward Folding Results in Simulation

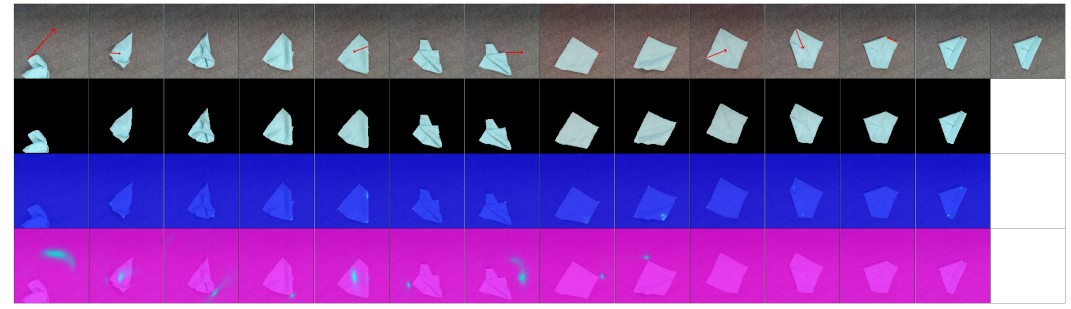

Figure 14: Qualitative Result of Corners-Edge Inward Folding in Reality

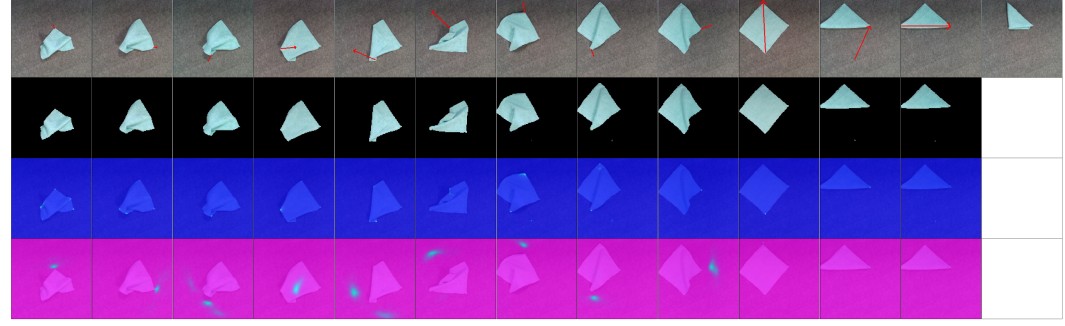

Figure 15: Qualitative Result of Diagonal Cross Folding in Reality

