# OpenReview forum: "JA-TN: Pick-and-Place Towel Shaping from Crumpled States based on TransporterNet with Joint-Probability Action Inference"
_robot-learning.org/CoRL/2024/Conference — CoRL 2024_

### Official Review · Reviewer_4Ctm · 2024-07-21
**Want to see real robot videos and photos**

**Originality:** 3
**Technical Quality:** 3
**Clarity Of Presentation:** 3
**Potential Impact:** 3
**Recommendation:** 3
**Confidence:** 4

**Review:**

Strength:
1. Tackles a difficult robotics problem
2. Shows positive results in the sim and claims successful sim to real transfer

Weakness:
1. It’s unclear what the contribution of the paper is. From line 93: The method of filtering only points on the cloth is an obvious choice one would make when applying an existing pick-and-place architecture for cloth folding. It doesn’t make sense to sample pick points not on the cloth. Similarly, changing batch size from 1 to 10 is hyper-parameter tuning and doesn’t count as a contribution.
2. I am not opposed to purely experimental papers and in fact, I encourage such papers. If indeed the paper is purely experimental, then it should be positioned as so with a very strong emphasis on real robot results. There should be learnings presented explaining details of which choices mattered the most in good policy performance on the real robot. In that way, the paper would add value to the robotics community allowing others to re-produce their approach and extend the learnings to similar domains.
2. Difficult to appreciate that there is a real robot involved from the photos and videos presented in the Appendix and supplementary material. The real robot results mentioned would be the strongest part of the paper and the authors should emphasize more on them with photos and a video and details on sim to real transfer choices

**Quality Of The Limitations Section:**

3

**Questions For Rebuttal:**

Please refer to the weakness above

**Robotics Focus:**

4

**Summary Of Paper:**

This paper presents a method to fold a piece of fabric from difficult initial states such as crumpled state. The method improves upon the Transporter Network by adding a mask around the piece of fabric and a tuning a few hyperparameters such as batch size. The authors show positive results in simulation and mention real robot transfer but real robot videos are missing in the supplementary material. It is difficult to see that there is a real robot from the photos in the paper (such as in Fig. 9).

**Summary Of Recommendation:**

The paper tackles a difficult robotics problem but I feel it needs restructuring and more experimental evidence. Post rebuttal edit: I think the paper looks much better now with the increased focus on real robot results. Hence, I now recommend accepting it.

---

### Official Review · Reviewer_WykJ · 2024-07-23
**Although the results show improvement, creating more benchmarks, oracle policies and providing more experiments maybe is recommended.**

**Originality:** 1
**Technical Quality:** 3
**Clarity Of Presentation:** 3
**Potential Impact:** 2
**Recommendation:** 3
**Confidence:** 4

**Review:**

## Strengths

The paper addresses an important problem and is of interest to the sub-community of non-rigid body manipulation. The expert primitives/policies designed can be widely used by the community for further data collection on different kinds of materials. In addition, improved inference performance shows that adding the mask helps the success rate. Extensive experiments are done on the 3 benchmarks proposed that show the effectiveness of the mask filter and joint probability for action prediction during inference. An ablation study justifies the use of the proposed two contributions.

## Weaknesses

The technical contribution is minor as it adds on top of existing works [1], [2].

1. A mask of the object is applied to the existing output of the TN network.
2. A single RGB-D image is used instead of 3D-reconstruction from multiple camera view images.
3. Training tricks like data augmentation.
The variety of objects experimented on is minimal.

[1] Deep imitation learning of sequential fabric smoothing from an algorithmic supervisor.
[2] Transporter networks: Rearranging the visual world for robotic manipulation

**Quality Of The Limitations Section:**

3

**Questions For Rebuttal:**

The reviewer suggests experimenting with more realistic non-rigid bodies or even a larger variety of non-rigid objects (ropes) and providing

1. oracle policies
2. benchmarks

for the same would make this a good contribution to the community.  However, this may require using a different physics simulator, which the reviewer understands may not be feasible.

The reviewer also recommends highlighting the paper's contributions to distinguish it from the related work.

**Robotics Focus:**

4

**Summary Of Paper:**

The paper addresses the problem of fabric folding from crumpled state on square and rectangular fabrics. The proposed method extends Transporter Networks, TN trained using behavior cloning (BC) predicts pick and place points directly from a 3D reconstruction of multi-view camera images. The extended version as proposed adds a mask to filter out the predicted pick points not on the object and uses the joint probability of filtered pick and place to determine both the pick - place actions during inference leading to improvement in performance. In addition, the paper introduces 3 benchmarks along with oracle expert policies. These policies behave as primitives (e.g., Revealing-corner, Revealing Edge) and are used to collect data for behavior cloning. Quantitative and Qualitative results are shown for simulation and real experiments.

**Summary Of Recommendation:**

Although the results show improvements, the contribution is minor. More oracle policies and benchmark design on garments and other non-rigid bodies would be of more interest to the community as a benchmarking and dataset paper.

---

### Official Review · Reviewer_zcen · 2024-07-23
**Sim2Real method for cloth manipulation with novel learning method and planner distillation**

**Originality:** 4
**Technical Quality:** 4
**Clarity Of Presentation:** 3
**Potential Impact:** 3
**Recommendation:** 4
**Confidence:** 4

**Review:**

## Summary

This paper introduces MJ-TN (Mask-Filtered Joint-Probability TransporterNet), a method for robotic towel manipulation that can flatten, fold from flattened states, and fold from crumpled states. The approach extends the TransporterNet architecture with mask-filtered joint-probability action inference to improve operational efficiency. The authors develop three benchmark domains based on SoftGym with towel-shaping tasks and corresponding oracle expert policies. They evaluate MJ-TN's performance against state-of-the-art controllers in simulation and demonstrate its ability to transfer to real-world scenarios. The method is tested on various types of fabrics and folding tasks, showing robust performance across different scenarios. The authors claim the following contributions:

* Development of MJ-TN, which achieves state-of-the-art performance for flattening, folding-from-flattened, and folding-from-crumpled tasks in simulation.
* Creation of three standard benchmark domains with towel-shaping tasks and corresponding Oracle expert policies.
* Demonstration of sim-to-real transfer capabilities for MJ-TN in physical trials.

### Strengths

Overall, the work is a nice contribution to the cloth manipulation literature. It makes good use of learning and provides demonstrations of a real system. In particular, the key strengths of the paper are:

* Comprehensive evaluation: The authors test their MJ-TN method across a wide range of scenarios, including different fabric types, sizes, and folding tasks. They also compare their results against state-of-the-art methods in both flattening and folding tasks, providing a thorough assessment of their approach's capabilities.

* Real-world evaluation: The paper demonstrates the method's ability to transfer from simulation to real-world scenarios, showing its potential for actual robotic applications. This sim-to-real transfer is a crucial step in bridging the gap between simulated environments and practical implementations.

### Weaknesses

* It is interesting to me that the authors are distilling a policy from a fully observed simulation to a partially observed robot deployment. It would be nice to see some discussion of this approach (in comparison to, e.g., RL)

* While the testing on the robot is quite good, further experiments in more diverse conditions would help to support the claims.

* It is a small point, but it would be good to see some evaluation of the Oracle policies that generate the data. Are there any configurations they struggle in?

**Quality Of The Limitations Section:**

2

**Questions For Rebuttal:**

I'm curious if the authors have any responses to my critiques above. I think there is room for improvement, however, I don't think there are any specific issues that need to be addressed prior to acceptance.

**Robotics Focus:**

4

**Summary Of Paper:**

See Review

**Summary Of Recommendation:**

Overall, this is a good demonstration of learning on a robotic platform that demonstrates strong performance on an interesting problem

---

### Author Rebuttal · Authors · 2024-08-13

We would like to thank the reviewers for their thorough and stimulating comments. We believe that the changes we made to the manuscript as a result of these reviews have resulted in a much stronger paper.

The rebuttal ZIP file contains the following:

1) A reworked manuscript, with changes highlighted in blue. This clarifies the experimental set-up requested by reviewer 4Ctm and the Area Chair XsBw (lines 182-202).  Figure 4 also explains our real-world setup in more detail and adds detail about the masking and image pre-processing needed for experiments (XsBw). The new draft also clarifies the need for demonstrations, which compares favourably to existing work (requested by XsBw, lines 211-229). There is also an extended appendix with more detail, including evaluation on non-rectangular fabrics in Figure 10 (WykJ), and more details on experimental set-up in Sec 7.3 (4Ctm).

2) The results reported in Table 6 include a human in the loop to focus on policy generation rather than grasping, but the picking actions were generated entirely by MJ-TN acting directly on live camera data. We clarified this in the reworked manuscript as requested by 4Ctm and WykJ. This hybrid approach eliminates issues relating to misgrasping and other grasping errors which are commonly reported in other papers in this domain. We believe that this meets the CoRL requirements to "provide convincing evidence that simulation experiments are transferable to real robots". However, we agree that full robot experiments are better, so:

3) We submit several videos showing the robot successfully flattening and folding garments in a fully autonomous fashion. These are accompanied by the step-by-step images in the same subfolder as additional evidence. See also Figure 4c in the updated manuscript for the photos requested by 4Ctm. We are currently working on a much larger-scale evaluation of autonomous folding on multiple robot platforms with partner institutions but we hope that these videos demonstrate that the method works in a fully autonomous setting with a real UR3e robot.

4) We have included the code for the oracle used to flatten and fold in simulation (WykJ). This is a novel contribution and an essential element of making the learning work.

5) We have included the updated code for MJ-TN, with a diff showing all the changes needed to make learning work in this domain (garment flattening and folding). This evidences significant updates to the TN code.

---

### Decision · Program_Chairs · 2024-09-04

**Decision:**

Accept

**Comment:**

This paper studies pick-and-place actions for towel manipulation by building on the Transporter Network architecture. The rebuttal convinced two reviewers who rated "weak reject" to increase scores to "weak accept." There is somewhat limited technical novelty here; for example the "mask-filtered" part should be de-emphasised; in fact this area chair strongly recommends removing that term from the paper title all together.

Nonetheless, the empirical results appear good enough for the robot learning community. The authors are encouraged to place more emphasis on describing their real world manipulation results. There has been some increased emphasis on the revised version from the rebuttal but there should be even more emphasis (for maximum impact).


Strengths:
- It tackles the important problem of shaping and adjusting deformable towels.
- Extending and improving Transporter Networks is impressive.

Weaknesses:
- The paper may be more experimental (instead of proposing a new algorithm) and thus it would help to have substantial real world robot experiments.
- The contributions may be limited compared to current work. For example, some of these might reduce to hyperparameter tuning (e.g., changing batch size). Also, filtering pick points so that they only go on the cloth might not be enough of a technical contribution.
- Changing colours of the fabric is interesting but also less interesting from a novelty and generalisation perspective, as a reasonable method should be robust to object color.
- It also requires a substantial amount of demonstrations.